# Two-Year Scale-Up of Seasonal Malaria Chemoprevention Reduced Malaria Morbidity among Children in the Health District of Koutiala, Mali

**DOI:** 10.3390/ijerph17186639

**Published:** 2020-09-11

**Authors:** Hamma Maiga, Jean Gaudart, Issaka Sagara, Modibo Diarra, Amadou Bamadio, Moussa Djimde, Samba Coumare, Boubou Sangare, Yeyia Dicko, Aly Tembely, Djibril Traore, Alassane Dicko, Estrella Lasry, Ogobara Doumbo, Abdoulaye A. Djimde

**Affiliations:** 1Institut National de Santé Publique, Bamako BP: 1771, Mali; hmaiga@icermali.org; 2Malaria Research and Training Center, Department of Epidemiology of Parasitic Diseases, Faculty of Medicine and Dentistry, Faculty of Pharmacy, University of Sciences, Techniques and Technologies of Bamako, Bamako BP: 1805, Mali; isagara@icermali.org (I.S.); modibod@icermali.org (M.D.); bamadio@icermali.org (A.B.); mdjimde@icermali.org (M.D.); scoumare@icermali.org (S.C.); boubous@yahoo.fr (B.S.); ydick2002@yahoo.fr (Y.D.); tembal2006@yahoo.fr (A.T.); dtraore@icermali.org (D.T.); adicko@icermali.org (A.D.); okd@icermali.org (O.D.); 3Aix Marseille Univ, APHM, IRD, INSERM, UMR1252 SESSTIM Sciences Economiques & Sociales de la Santé & Traitement de l’Information Médicale, Hop Timone, BioSTIC, Biostatistic & ICT, 13385 Marseille, France; jean.gaudart@univ-amu.fr; 4Médecins Sans Frontières (MSF), New York, NY 10006, USA; estrella.lasry@newyork.msf.org

**Keywords:** malaria, morbidity, seasonal malaria chemoprevention, Mali

## Abstract

Background: Previous controlled studies demonstrated seasonal malaria chemoprevention (SMC) reduces malaria morbidity by >80% in children aged 3–59 months. Here, we assessed malaria morbidity after large-scale SMC implementation during a pilot campaign in the health district of Koutiala, Mali. Methods: Starting in August 2012, children received three rounds of SMC with sulfadoxine-pyrimethamine (SP) and amodiaquine (AQ). From July 2013 onward, children received four rounds of SMC. Prevalence of malaria infection, clinical malaria and anemia were assessed during two cross-sectional surveys conducted in August 2012 and June 2014. Investigations involved 20 randomly selected clusters in 2012 against 10 clusters in 2014. Results: Overall, 662 children were included in 2012, and 670 in 2014. Children in 2014 versus those surveyed in 2012 showed reduced proportions of malaria infection (12.4% in 2014 versus 28.7% in 2012 (*p* = 0.001)), clinical malaria (0.3% versus 4.2%, respectively (*p* < 0.001)), and anemia (50.1% versus 67.4%, respectively (*p* = 0.001)). A propensity score approach that accounts for environmental differences showed that SMC conveyed a significant protective effect against malaria infection (IR = 0.01, 95% CI (0.0001; 0.09), clinical malaria (OR = 0.25, 95% CI (0.06; 0.85)), and hemoglobin concentration (β = 1.3, 95% CI (0.69; 1.96)) in 2012 and 2014, respectively. Conclusion: SMC significantly reduced frequency of malaria infection, clinical malaria and anemia two years after SMC scale-up in Koutiala.

## 1. Introduction

Malaria remains a leading cause of morbidity and mortality, causing an estimated 228 million cases of clinical malaria and 405 thousand deaths worldwide [1]. More than 93% of malaria cases and 94% of malaria deaths occur in the African region. Children under 5 years of age are the most vulnerable group affected, accounting for 67% of all malaria deaths worldwide [1]. Vector control has substantially reduced mortality and morbidity from malaria [2], but in high transmission settings, these interventions provide only partial protection. The rapid spread of insecticide and mosquito behavioral resistance compromises these current control measures, as failures of vector control interventions, such as indoor residual spraying (IRS) and insecticide-treated nets (ITN), are being reported [3]. It is important to consider additional alternative measures that could be used to control malaria in areas where infection is highly seasonal [3].

In 2012, the World Health Organization (WHO) recommended seasonal malaria chemoprevention (SMC) with sulfadoxine-pyrimethamine and amodiaquine (SP plus AQ) in the Sahel countries in Africa to reduce malaria among children under 5 years of age [4]. Seasonal malaria chemoprevention (SMC) is the administration of treatment doses of longer-acting antimalarial medications at monthly intervals in areas of exclusively seasonal transmission [4]. The goal is to treat any existing infections and maintain protective drug concentrations in the blood throughout a complete transmission season [4].

SMC is a strategy to prevent malaria in children under 5 years of age across the Sahel sub-region, where most childhood malaria mortality and morbidity occurs during a few months (approximately 4 months). SMC has been associated with a reduction in the incidence of malaria infection, as well as uncomplicated and severe malaria, in the range of 67% to 86% in children [5,6,7]. SMC is cost-effective [8], and a high level of coverage can be obtained when delivered by community health workers under close supervision of healthcare professionals [9]. Several controlled studies have shown that the intervention was well-tolerated [5,7,10,11,12].

No significant rebound in clinical malaria incidence was observed in the year after SMC was given during initial studies conducted in Senegal, Ghana, and Mali [5,6,7,11]. Large-scale SMC implementation in Mali began in 2012 in the district of Koutiala, through a collaboration between Médecins Sans Frontières (MSF) and the National Malaria Control Program of Mali (NMCP).

Malaria is the primary cause of morbidity and mortality in Mali [13,14], particularly among children under five years of age. It is also the leading cause (35%) of pediatric health center consultations and hospitalization (45%) [15]. The parasite index varies from 10% in certain regions of the Sahel to 80% in the South (Savannah zone) [16]. *Plasmodium falciparum* is the predominant species, with approximately greater than 85.9% of the parasite formula, followed by *Plasmodium malariae* (10–15%) and *Plasmodium ovale* 1% [16].

Our primary observations of SMC showed an increasing prevalence of two resistance markers, *Plasmodium falciparum* dihydropteroate synthase (*Pf*dhps) 540E and *P. falciparum* dihydrofolate reductase (*Pf*dhfr)-dhps quintuple mutant genotype, in the treated population several months after the last SMC round in 2014 [17]. These very high levels of triple, quadruple, and quintuple mutants could compromise the effectiveness of SP, but there are no data linking the number of mutations to in vivo resistance against SP plus AQ.

The present study assessed the impact of this intervention on prevalence of malaria infection, clinical malaria, and anemia among children aged 3–59 months in the health district of Koutiala, Mali.

## 2. Materials and Methods

Study site: The study was conducted in community health areas located in the district of Koutiala, a region of Sikasso in the south of Mali, with an average rainfall of about 700 mm per year. Koutiala is located at about 420 km south of Bamako, the capital city, with about 575,000 inhabitants in 2012, and 580,453 inhabitants in 2014, according to the National Institute of Statistics of Mali [18]. Koutiala is divided into urban and rural areas. Typically, as defined by the United Nations, the population living in towns of 2000 or more in national and provincial capitals is classified as urban, while rural indicates a geographic area that is located outside towns and cities [19]. Malaria transmission in Koutiala is known to be highly seasonal, with >80% of malaria cases occurring during the rainy season [20].

### Study Design and Participants

SMC delivery: MSF, in collaboration with the NMCP, delivered SMC in 2012 and 2013. The delivery teams used a combination of fixed post and door-to-door strategies for all children aged 3–59 months. The investigators on this study were not involved in the delivery of SMC drugs.

Cross-sectional study: Two cross-sectional surveys were conducted, first in August 2012 (a few days before the start of SMC implementation), and second in June 2014 (~8 months after the last doses of SMC drugs were distributed). After identification of eligible children, informed consent was obtained from their parents prior to their interview and inclusion. Children were eligible to participate to the study if they were aged 3–59 months at the time of enrolment, and residents of the study area. Children with the following criteria were excluded: severe or chronic illness, such as acquired immune deficiency syndrome (AIDS), and known adverse reaction to SP or AQ. During the 2012 malaria transmission season, three cycles of SMC with SP plus AQ were provided in August, September, and October. During the 2013 malaria transmission season, four cycles of SMC with SP plus AQ were provided in July, August, September, and October.

The three primary endpoints were (i) prevalence of malaria infection, defined as the presence of asexual stage of *Plasmodium falciparum* malaria (by microscopy) without clinical sign; (ii) prevalence of clinical malaria, defined as the presence of asexual stage of *P. falciparum* (by microscopy), plus fever or history of fever; and (iii) proportion with anemia (as measured by hemoglobin concentration <11 g/dL) [21], before and after the intervention period. For all participants, a thick blood smear and dried blood spots were collected by finger prick. Axillary temperature (≥37.5 °C was considered fever) and history of fever within the previous 24 h was assessed, a rapid malaria diagnostic test (RDT Standard Diagnostics (SD) BIOLINE Malaria Ag P.f^®^, (BIOLINE, Suwon City, South Korea) was conducted, and anemia was measured for all enrolled children. The participants included both symptomatic and asymptomatic children, provided they were aged 3–59 months during the survey. Symptomatic cases were diagnosed with RDT; in case of positivity, they received artemether-lumefantrine (Coartem^R^), per National Malaria Treatment Guidelines (NMTG) of Mali, and those positive did not receive SMC.

Laboratory methods: Thick blood films were air dried, stained with 3% Giemsa, and examined for malaria parasites by two well-trained technicians; 100 high-power fields under an oil immersion objective were counted before a film was declared negative. Parasite density was determined by counting the number of parasites present per white blood cell (WBC) on a thick smear, assuming a WBC count of 8000 per microliter. In the case of a discrepancy (positive/negative or a difference in parasite density greater than 30%), a third reading was done. Hemoglobin concentrations were measured using a hemoglobin analyzer (HemoCue HB 301, Angelholm, Sweden) on blood obtained by finger prick.

Data management and statistical analysis: Data were collected on standardized forms, double-entered, and verified using MS Access and then exported to R Statistical Software (version 3.0.3) for additional cleaning and analysis. Proportions of children with malaria infection, clinical malaria, and anemia were first compared using a univariate approach with chi-square or Fisher’s exact tests per age category (3–11 months versus 12–59 months) and area (rural vs. urban). Because exposure to malaria varies due to differences in location, meteorology, and environmental characteristics, a propensity score approach was implemented to reduce confounding biases [22]. Propensity score estimation was measured between baseline vs. post-intervention for malaria infection, clinical malaria, and hemoglobin concentration. Propensity score estimation was based on village location (longitude and latitude), sex, age, population count, urban or rural characteristic, rain, temperature, humidity, wind speed, and cumulative number of malaria cases from January to the survey date. This approach accounts for different environmental characteristics between villages and survey years, to correct for exposure differences between survey months. For binomial outcomes, a weighted logistic regression was used to count data outcome (malaria infection). For multivariate analysis (clinical malaria) a weighted quasi-Poisson regression was used to account for over-dispersion; and for continuous outcomes (hemoglobin concentration), a weighted linear regression was used. All regression models were weighted by the propensity score. Statistical significance was set to *p* < 0.05.

Assessment of malaria morbidity and sample size: A random sample of 20 clusters (villages/neighborhoods where the community health facilities were located) was selected from all 42 enumerations (villages/neighborhoods where community health facilities are located) in the health district of Koutiala. The 20 clusters were selected on a proportional basis after stratification of urban and rural communes according to the target population of each stratum. In each cluster, at least 32 children were surveyed, and in each household, all target children were sampled. All villages outside county boundaries were considered rural areas, while urban areas were represented by Koutiala city neighborhoods. Each team was accompanied by a representative of the village chief or district, to identify the consensual center of the village/neighborhood. Random selection of households was conducted by flipping a pen, then using its tip to indicate the direction to follow for the investigation. Once a direction was chosen, the investigation began, and focused on families on the right side of the street up to the end, then the opposite side. This process was repeated until the sample size of households was achieved. For each family, a legal guardian of a child volunteer aged 3–59 months provided informed consent, and was interviewed via questionnaire. A total of 20 villages and neighborhoods yielded a sample size of 640 children, sufficient to compare primary parameters before and after SMC implementation. For logistical reasons, in 2014, the number of villages/neighborhoods was changed to 10, from which 64 children were surveyed per cluster. All participants received an initial history and physical examination, followed by blood smear and hemoglobin measurement collected by finger prick. Uncomplicated and severe malaria were also documented for all volunteers at each health center, before, during, and after SMC implementation from 2011 to 2014. The cumulative number of malaria cases was used in the propensity score from January to the survey date. Meteorology and environmental characteristics (rain, temperature, humidity, and wind speed) were collected at each center.

Ethics clearance: The study protocol was reviewed and approved by the Ethical Committee of the Faculty of Medicine and Dentistry and Faculty of Pharmacy/University of Sciences, Techniques and Technologies of Bamako (USTTB), Mali (N°2015/109/CE/FMPOS). Community clearance was obtained after meetings with community authorities, leaders, heads of families, and other community members of each locality prior to the start of the study. Individual, written, informed consent was obtained from a parent or guardian of each child prior to screening. The study was conducted according to current Good Clinical Practices (cGCP) by investigators from Malaria Research and Training Center (MRTC), Bamako, Mali.

## 3. Results

### 3.1. Demographic Characteristics of the Study Participants at Baseline and Post-Intervention SMC Implementation

From 4 to 9 August 2012 and 20 to 30 June 2014, 662 and 670 children aged from 3 to 59 months were enrolled in the study, respectively. The proportion of children aged 3–11 months were 11.4% (N = 662) and 10.3% (N = 670) at baseline and post-intervention, respectively (Table 1). The proportion of fever or history of fever was 38.7% (256) at baseline, and 17.9% (120) post-intervention. The proportion of malaria infection was 28.7% (190) at baseline, and 12.4% (83) post-intervention; *p* = 0.001. The proportion of all children with clinical malaria was 4.2% (28) at baseline vs. 0.3% (2) post-intervention; *p* < 0.001. The anemia prevalence was 67.4% (446) at baseline, and 50.1% (336) post-intervention; *p* = 0.001.

### 3.2. Multivariate Regression Analysis of Pre–Post-SMC of Malaria Infection, Clinical Malaria and Anaemia

SMC treatment conferred significant reduction in malaria infection, with an incidence ratio of IR = 0.01; 95% CI (0.0001, 0.09), using a weighted (propensity score) quasi-Poisson regression (Table 2). Similarly, SMC treatment resulted in significant reduction in clinical malaria (OR = 0.25; 95% CI (0.06; 0.85)), using a weighted (propensity score) binomial regression. Hemoglobin concentration was also corrected by SMC (β = 1.3; 95% CI (0.69; 1.96)), using a weighted (propensity score) linear regression.

### 3.3. Prevalence of Sexual and Asexual Stage of P. falciparum and Anaemia at SMC Baseline Per Age Category and Area

The proportion of asexual stage *P. falciparum* malaria was lower among children aged 3–11 months than among children aged 12–59 months (5.3% vs. 32.8% respectively; *p* < 0.001; Table 3). The proportion of asexual stage *P. falciparum* was higher in the rural area (34.7%) compared to the urban area (2.9%) (*p* < 0.001). No case of severe malaria was observed during this study (table not shown). The proportion of children with anemia (Hb < 11 g/dL) was significantly higher in the rural area, compared to the urban area (69.7% vs. 55.3%; *p* = 0.004). Anemia was also more prevalent among children age of 3–11 months, compared to children aged 12–59 months (79.0% vs. 66.0%: *p* = 0.02). The difference was larger for moderate anemia (Hb < 8 g/dL), with a proportion of 10.8% vs. 1.9% in the rural and urban area, respectively (*p* = 0.005). Moderate anemia prevalence was lower among the children aged 3–11 months vs. 12–59 months (*p* = 0.03). A total of four cases of severe anemia (Hb < =5 g/dL) occurred in rural areas among children aged 12–59 months (*p* = 0.39). No case of severe anemia was observed in urban areas, or in children aged 3–11 months.

### 3.4. Prevalence of Sexual and Asexual Stage of P. falciparum and Anemia Per Age Category and Area in SMC Post-Intervention

The proportion of asexual stage *P. falciparum* was lower among children aged 3–11 months than among those aged 12–59 months (2.7% vs. 13.6% respectively; *p* = 0.004; Table 4). The proportion of children with anemia (Hb < 11 g/dL) was significantly higher among children aged 3–11 months, compared to children aged 12–59 months (69.9% vs. 47.7%; *p* < 0.001). The proportion of anemic children was significantly higher in rural areas compared to urban areas (52.7% vs. 43.9%, respectively; *p* = 0.04). No difference was observed for moderate anemia (Hb < 8 g/dL) and severe anemia (Hb ≤ 5 g/dL) (*p* > 0.5). No case of severe malaria was observed during this study (table not shown).

## 4. Discussion

Previous studies determined that during the highest malaria transmission season, SMC provides substantial additional protection against episodes of clinical malaria and severe malaria in children sleeping under long-lasting insecticide-treated bed nets [6,7,23,24,25]. This study demonstrates the impact of SMC on malaria morbidity, in terms of malaria infection, clinical malaria, and anemia, after two years of large-scale implementation in Mali.

In our study, the proportion of malaria infection, clinical malaria, and anemia were significantly decreased after two years of pilot implementation of SMC in the health district of Koutiala, Mali. The national survey of parasitemia (led by the National Malaria Control Program (NMCP) with the support of USAID President’s Malaria Initiative (PMI)) among children 6–59 months of age showed an overall parasitemia of 59.2% in the Sikasso region in 2010, with 52% of children aged 6–59 months infected with malaria at the time of interview in Mali [26], versus 28.7% and 12.4% in the district of Koutiala in August 2012 and June 2014, respectively. The results of the national survey according to the place of residence indicated a prevalence of malaria 3.5 times higher among children in rural areas than in urban areas in Africa [19]. Age is a significant risk factor for the prevalence and density of parasitemia [27]. One of the main determinants of the age distribution with regards to morbidity is the development of anti-parasite immunity that restricts the density of asexual parasitemia [28].

Parasitemia peaks in children less than 5 years old, and subsequently declines in an age-dependent manner [28]. Therefore, a significant association between age and parasite density among children aged between 6 months and 6 years has been shown, with younger children more likely to have a density above 5000 parasites/μL [29]. In our study, the proportion of asexual stage *P. falciparum* was lower in children aged 3–11 months, compared to 12–59 months. This may be due to the fact that the risk of clinical event increases from birth to about 6 months of age, depending on the transmission rate, and begins at around 3 to 4 months of age [28].

The proportion of anemia within the national survey [26] showed an overall anemia count of 91.7% in the Sikasso general population in 2010, and 82% of children 6–59 months were anemic in 2013, against 67.4% and 50.1% in the district of Koutiala in 2012 and 2014, respectively. With regards to moderate-to-severe anemia, the national survey in 2010 found a prevalence of 34.8% in the region of Sikasso, against 9.4% and 1.6% in the district of Koutiala in 2012 and 2014, respectively. In our study, anemia affected more children in rural areas than in urban areas (70% against 55%). It also should be noted that 11% of rural children suffered from moderate anemia, and 2% suffered from severe anemia. Some features of clinical malaria (e.g., anemia) become less severe with age [30]. As such, anemia was significantly higher in children aged 3–11 months (79%), compared to children aged 12–59 months (66%).

Limitations of this population study include (1) the fact that surveys were conducted during different periods (months); and (2) the lack of a comparison region, as SMC was implemented in all neighboring health districts of Koutiala. In 2012, study parameters were measured in August, while in 2014, these were measured in June. This difference in period could affect the parameter comparisons between baseline and post-intervention. The propensity score approach was used to reduce the impact of this confounding bias, rendering the groups comparable. Another limitation of the study is related to the fact that the cross-sectional study design limits the level of evidence collected from the study. Others confounding factors were not accounted for, such as parental behaviors (i.e., using insecticide-treated nets (ITNs), long-lasting insecticide-treated nets (LLINs), or fumigation), and genetic make-up of the children could account for some of the differences observed in the pre- and post-analysis.

Our results are consistent with previous studies that found a substantial reduction in malaria infection, clinical malaria, and anemia [6,7,10,23,24,25,31,32,33,34]. In Ghana, there was no significant increase in the incidence of clinical malaria in the post-intervention period in one-year-old children that received SMC [6] Another study from Mali has shown that SMC implementation was associated with reduction in all-cause mortality and in hospital admissions [35]. These data support our findings about SMC in Mali. This study includes the retrospective assessment of outcomes that could be subject to recall bias, but our study is a prospective study. These two findings support a wide implementation of the strategy to reduce malaria burden in Sahelian countries.

The current study differs from those prior since this was an observational study, and not a controlled trial. Nevertheless, our study shows that SMC significantly reduces malaria morbidity (infection, clinical malaria, and anemia). SMC implementation in our study was also associated with insecticide-treated nets (ITNs), long-lasting insecticide-treated nets (LLINs), rapid diagnostic tests (RDTs), artemisinin-based combinations (ACTs), and a campaign of sensibility. These methods existed previously, but were not implemented at the population level. These interventions may also have reduced malaria morbidity along with SMC. Several sets of experimental data support the hypothesis [36] that interventions that reduce *P. falciparum* transmission intensity will reduce high-density parasitemia and malaria-associated morbidity and mortality. Therefore, ITNs provide protection against morbidity and mortality attributable to malaria [37,38], as well as the risk of fever [39] and the prevalence or incidence of parasitemia [40].

## 5. Conclusions

Malaria infection, clinical malaria, and anemia prevalence were significantly reduced after two years of large-scale implementation of SMC, together with LLIN, RDT, and ACT strategies, in the health district of Koutiala.

## Figures and Tables

**Table 1 ijerph-17-06639-t001:** Demographic, clinical, and laboratory participant characteristics before and post-seasonal malaria chemoprevention (SMC) intervention.

	Baseline		Post-Intervention		
	N = 662		N = 670		
	n	(%)	n	(%)	*p*-Value *
Sex ratio (female)	361	(54.5)	333	(49.7)	-
Age (3–11 months)	75	(11.3)	73	(10.9)	-
Urban	192	(29.0)	190	(28.4)	-
Fever or history of fever	256	(38.7)	120	(17.9)	0.001
Malaria infection	190	(28.7)	83	(12.4)	0.001
*Plasmodium malariae*	2	(0.3)	3	(0.4)	0.99
*Plasmodium ovale*	3	(0.5)	0	(0)	0.24
Gametocyte of *Plasmodium falciparum*	60	(9.1)	15	(2.2)	0.001
Clinical malaria	28	(4.2)	2	(0.3)	<0.001
Hemoglobin concentration (Hb <11 g/dL)	446	(67.4)	336	(50.1)	0.001

* *p*-value for univariate analysis.

**Table 2 ijerph-17-06639-t002:** Prevalence of malaria infection, clinical malaria, and anemia, before and post-SMC intervention.

	Before SMC		After SMC			
	N = 662		N = 670			
	n	(%)	n	(%)	Impact Measure 95% CI	*p*-Value **
Malaria infection	190	(28.7)	83	(12.4)	0.01 [0.0001; 0.09] §	0.02
Clinical malaria	28	(4.2)	2	(0.3)	0.25 [0.06; 0.85] §§	0.03
Hemoglobin concentration (Hb <11 g/dL)	446	(67.4)	336	(50.1)	1.3 [0.69; 1.96] §§§	< 0.01

§ Incidence Ratio; §§ Odd Ratio; §§§ β coefficient; ** *p*-value for multivariate analysis.

**Table 3 ijerph-17-06639-t003:** Prevalence of sexual and asexual stage of *Plasmodium falciparum* and anemia in SMC baseline per age category and residence.

	Age (Month)			Area		
	3–11 N = 75 %	12–59 n = 585 %	*p*-Value *	Urban n = 103 %	Rural n = 557 %	*p*-Value ***
Asexual-stage parasites	5.3	32.8	<0.001	2.9	34.7	<0.001
Sexual-stage parasites	1.3	10.1	0.01	0.0	10.8	0.001
Hb < 11 g/dL	79.0	66.0	0.02	55.3	69.7	0.004
Hb < 8 g/dL	2.6	10.3	0.03	1.9	10.8	0.005
Hb < = 5 g/dL	0.0	0.7	0.50	0.0	0.7	0.390

* *p*-value for univariate analysis.

**Table 4 ijerph-17-06639-t004:** Prevalence of sexual and asexual stage of *P. falciparum* and anemia in SMC post-intervention per age category and residence.

	Age (Month)			Area		
	3–11 n= 73 (%)	12–59 n = 597 (%)	*p*-Value *	Urban n = 190 (%)	Rural n = 480 (%)	*p*-Value *
Asexual-stage parasites	2.7	13.6	0.004	1.1	16.9	<0.001
Sexual-stage parasites	0.0	2.5	0.17	0.0	3.5	0.06
Hb < 11 g/dL	69.9	47.7	<0.001	43.9	52.7	0.04
Hb < 8 g/dL	1.4	1.7	0.85	1.6	1.7	0.97
Hb ≤ 5 g/dL	0.0	0.0	-	0.0	0.0	-

* *p*-value for univariate analysis.

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
