# Peer review of "Two-Year Scale-Up of Seasonal Malaria Chemoprevention Reduced Malaria Morbidity among Children in the Health District of Koutiala, Mali"

_ijerph, 2020, doi:10.3390/ijerph17186639_

Round 1

Reviewer 1 Report

Generally looking, the manuscript provides a fair assessment on the malaria morbidity after the SMC implementation in Koutiala district, Mali, using sound methodology. However, the originality and contributed values cannot be validated, so I think the paper does not meet the standard to be published in IJERPH. Here are some remarks that the author can improve upon:

  1. The study’s objectives and contributed values are not comprehensively disclosed. Moreover, the current objective is irrational because it lacks an explanation of why the assessment needs to be done.
  2. There should be a thorough explanation of how the assessment is done.
  3. Discussion is confusing and misleading. I agree that comparing the current study’ findings with findings of prior studies and the national program is a good idea, but the comparison is short of purposes. It is also unclear whether the results in the Discussion are findings of the current study or the national survey.
  4. Are there any policy implications?

Author Response

Dear Reviewer,

Thanks

Reviewer 2 Report

The study “Two years of scaling up seasonal malaria chemoprevention reduced malaria morbidity among children in the health district of Koutiala, Mali” will significantly advance the prevention of malaria. The study is important because.

Malaria is one of the tropical diseases in Africa that continues to claim the lives of children on the continent. I believe this study is critical to help curb some of these preventable diseases.

However, my enthusiasm for this study was significantly diminished by the writing style of the authors. The authors need to go back to make certain their ideas flow together, rephrase some of the sentences, and correct grammatical errors. Currently, it is very difficult to follow authors ideas without

Examples of those concerns are as follows

  1. The authors stated, “Malaria remains a leading cause of morbidity and mortality, causing an estimated 228 million cases of clinical malaria and 405 thousand deaths in 2019” The question is are those figures worldwide or only in Africa? The authors need to specify.
  2. The authors stated “More than 93% of malaria cases and 94% of malaria deaths occur in the World Health Organization (WHO) African Region, where the children aged under 5 years are the most vulnerable group affected, accounting for 67% of all malaria deaths worldwide” First, I am not sure what the authors mean by “… the World Health Organization (WHO) African region. If they are referring to the African continent, they should say the African continent or Africa. I think the World Health Organization (WHO) is not necessary. Second, the authors continued “…where the children aged under 5 years are the most vulnerable group affected, accounting for 67% of all malaria deaths worldwide” Something is missing in the whole sentence. I think breaking down the sentence into two different sentences will bring out the idea clearer or rephrasing the whole sentence will help your readers to understand the point.
  3. The authors stated, “Vector control has substantially reduced mortality and morbidity from malaria [2], but in high transmission settings, these interventions provide only partial protection and additional control measures are needed” Again, the authors referred to “vector control has” but the subsequent phrase referred to it as “these interventions” It could be “Vector control interventions or programs….”
  4. The authors stated, “A large part of this age group comprised of our group of children less than 1 year of age (about 6 months).” I am not sure what the authors mean by this sentence. It seems to me that this sentence is out of place or something is missing.
  5. The authors stated, “It also should be noted that 11% of rural children with moderate anaemia and 2% suffer from anaemia in the severe form.” This sentence, for instance, is incomplete. Should it be “It also should be noted that 11% of rural children suffer from moderate anaemia and an additional 2% of them suffer from the severe form of anaemia?” or something in that like that.

The above concerns are just a few examples of some issues that need to be addressed throughout the whole manuscript.

Another concern is making causal inferences from the results is stretching the study’s findings too far. I am very much aware that causal inferences can be deduced when you conduct propensity score estimation. However, my concern for the current study is that some confounding factors such as parental behaviors (i.e. using misquote repellents or misquote nets) and genetic make-up of the child could account for some of the

Author Response

Dear Reviewer,

Thanks

Reviewer 3 Report

Very interesting and well constructed article

Check the institutions of the authors, they are strangely indicated

Abstract: If we have the 95% CI we don't need the p, it gives twice the same information.

Introduction: it would be nice to have a little more information on Seasonal Malaria Chemoprevention

A little more information is needed on the epidemiological situation of malaria in Mali, burden of disease, prevalence etc.

Method :

Line 65: to be referenced in the literature

Line 123: Why 20 clusters? What is the methodological explanation?

Results :

Table 2: For the multivariate analysis I did not understand which variables were included in the model?

Another limitation of the study is simply related to the fact that since the study design is only cross sectional it limits the level of evidence of the study.

Author Response

Dear Reviewer,

Thanks

Round 2

Reviewer 1 Report

The manuscript quality has been improved significantly after revision, but I need to admit that I am not completely convinced by the scientific novelty of the current study. Some of the reasons are: 1) rather similar study has been conducted in Mali with almost similar age range (children under 5), and 2) the current study's data are seemingly outdated compared to that study. Please refer to:https://malariajournal.biomedcentral.com/articles/10.1186/s12936-020-03175-y

Still, the current study has several novel points compared to the study I mentioned above, so I recommend the authors to carefully compare the similarities and differences between the two studies and explain how the two studies' findings can supplement each other when rationalizing the study's objective and discussing the policy implication and scientific contribution. It should be noted that geographical difference is not a novel contribution.

Author Response

Dear Reviewer

Please the attachment

Thanks

Reviewer 2 Report

I think the authors ignore my major concern about making causal inferences from propensity score estimation without controlling for obvious confounder. Below is my initial concern.

Another concern is making causal inferences from the results is stretching the study’s findings too far. I am very much aware that causal inferences can be deduced when you conduct propensity score estimation. However, my concern for the current study is that some confounding factors such as parental behaviors (i.e. using misquote repellents or misquote nets) and genetic make-up of the child could account for some of the differences observed in the pre and post-analysis. These confounders were not accounted for so this is a limitation, that should be acknowledged.

Also, the authors refusal to track all the changes they made, makes it difficult to know where the changes occur. It would have been helpful had the authors tracked the changes they made.

Author Response

Dear Reviewer

Thanks

This manuscript is a resubmission of an earlier submission. The following is a list of the peer review reports and author responses from that submission.